# LSDA: Large Scale Detection through Adaptation

**Judy Hoffman$^\diamond$, Sergio Guadarrama$^\diamond$, Eric Tzeng$^\diamond$, Ronghang Hu$^\nabla$, Jeff Donahue$^\diamond$,**
$^\diamond$EECS, UC Berkeley, $^\nabla$EE, Tsinghua University
{jhoffman, sguada, tzeng, jdonahue}@eecs.berkeley.edu
hrh11@mails.tsinghua.edu.cn

**Ross Girshick$^\diamond$, Trevor Darrell$^\diamond$, Kate Saenko$^\triangle$**
$^\diamond$EECS, UC Berkeley, $^\triangle$CS, UMass Lowell
{rbg, trevor}@eecs.berkeley.edu, saenko@cs.uml.edu

## Abstract

A major challenge in scaling object detection is the difficulty of obtaining labeled images for large numbers of categories. Recently, deep convolutional neural networks (CNNs) have emerged as clear winners on object classification benchmarks, in part due to training with 1.2M+ labeled classification images. Unfortunately, only a small fraction of those labels are available for the detection task. It is much cheaper and easier to collect large quantities of image-level labels from search engines than it is to collect detection data and label it with precise bounding boxes. In this paper, we propose Large Scale Detection through Adaptation (LSDA), an algorithm which learns the difference between the two tasks and transfers this knowledge to classifiers for categories without bounding box annotated data, turning them into detectors. Our method has the potential to enable detection for the tens of thousands of categories that lack bounding box annotations, yet have plenty of classification data. Evaluation on the ImageNet LSVRC-2013 detection challenge demonstrates the efficacy of our approach. This algorithm enables us to produce a >7.6K detector by using available classification data from leaf nodes in the ImageNet tree. We additionally demonstrate how to modify our architecture to produce a fast detector (running at 2fps for the 7.6K detector). Models and software are available at lsda.berkeleyvision.org.

## 1 Introduction

Both classification and detection are key visual recognition challenges, though historically very different architectures have been deployed for each. Recently, the R-CNN model [1] showed how to adapt an ImageNet classifier into a detector, but required bounding box data for all categories. We ask, is there something generic in the transformation from classification to detection that can be learned on a subset of categories and then transferred to other classifiers?

One of the fundamental challenges in training object detection systems is the need to collect a large of amount of images with bounding box annotations. The introduction of detection challenge datasets, such as PASCAL VOC [2], have propelled progress by providing the research community a dataset with enough fully annotated images to train competitive models although only for 20 classes. Even though the more recent ImageNet detection challenge dataset [3] has extended the set of annotated images, it only contains data for 200 categories. As we look forward towards the goal of scaling our systems to human-level category detection, it becomes impractical to collect a large quantity of bounding box labels for tens or hundreds of thousands of categories.

---

$^*$This work was supported in part by DARPA's MSEE and SMISC programs, by NSF awards IIS-1427425, and IIS-1212798, IIS-1116411, and by support from Toyota.

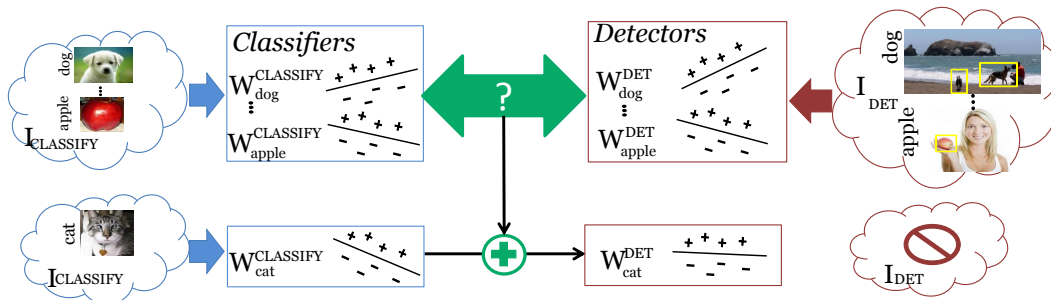

Figure 1: The core idea is that we can learn detectors (weights) from labeled classification data (left), for a wide range of classes. For some of these classes (top) we also have detection labels (right), and can learn detectors. But what can we do about the classes with classification data but no detection data (bottom)? Can we learn something from the paired relationships for the classes for which we have both classifiers and detectors, and transfer that to the classifier at the bottom to make it into a detector?

In contrast, image-level annotation is comparatively easy to acquire. The prevalence of image tags allows search engines to quickly produce a set of images that have some correspondence to any particular category. ImageNet [3], for example, has made use of these search results in combination with manual outlier detection to produce a large classification dataset comprised of over 20,000 categories. While this data can be effectively used to train object classifier models, it lacks the supervised annotations needed to train state-of-the-art detectors.

In this work, we propose Large Scale Detection through Adaptation (LSDA), an algorithm that learns to transform an image classifier into an object detector. To accomplish this goal, we use supervised convolutional neural networks (CNNs), which have recently been shown to perform well both for image classification [4] and object detection [1, 5]. We cast the task as a domain adaptation problem, considering the data used to train classifiers (images with category labels) as our source domain, and the data used to train detectors (images with bounding boxes and category labels) as our target domain. We then seek to find a general transformation from the source domain to the target domain, that can be applied to any image classifier to adapt it into a object detector (see Figure 1).

Girshick et al. (R-CNN) [1] demonstrated that adaptation, in the form of fine-tuning, is very important for transferring deep features from classification to detection and partially inspired our approach. However, the R-CNN algorithm uses classification data only to pre-train a deep network and then requires a large number of bounding boxes to train each detection category.

Our LSDA algorithm uses image classification data to train strong classifiers and requires detection bounding box labeled data for only a small subset of the final detection categories and much less time. It uses the classes labeled with both classification and detection labels to learn a transformation of the classification network into a detection network. It then applies this transformation to adapt classifiers for categories without any bounding box annotated data into detectors.

Our experiments on the ImageNet detection task show significant improvement (+50% relative mAP) over a baseline of just using raw classifier weights on object proposal regions. One can adapt any ImageNet-trained classifier into a detector using our approach, whether or not there are corresponding detection labels for that class.

## 2  Related Work

Recently, Multiple Instance Learning (MIL) has been used for training detectors using weak labels, i.e. images with category labels but not bounding box labels. The MIL paradigm estimates latent labels of examples in positive training bags, where each positive bag is known to contain at least one positive example. Ali et al. [6] constructs positive bags from all object proposal regions in a weakly labeled image that is known to contain the object, and uses a version of MIL to learn an object detector. A similar method [7] learns detectors from PASCAL VOC images without bounding box

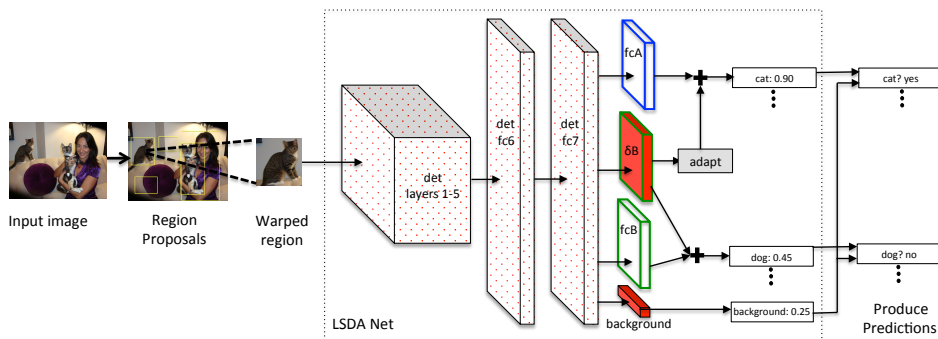

Figure 2: Detection with the LSDA network. Given an image, extract region proposals, reshape the regions to fit into the network size and finally produce detection scores per category for the region. Layers with red dots/fill indicate they have been modified/learned during fine-tuning with available bounding box annotated data.

labels. MIL-based methods are a promising approach that is complimentary to ours. They have not yet been evaluated on the large-scale ImageNet detection challenge to allow for direct comparison.

Deep convolutional neural networks (CNNs) have emerged as state of the art on popular object classification benchmarks (ILSVRC, MNIST) [4]. In fact, "deep features" extracted from CNNs trained on the object classification task are also state of the art on other tasks, e.g., subcategory classification, scene classification, domain adaptation [8] and even image matching [9]. Unlike the previously dominant features (SIFT [10], HOG [11]), deep CNN features can be learned for each specific task, but only if sufficient labeled training data are available. R-CNN [1] showed that fine-tuning deep features on a large amount of bounding box labeled data significantly improves detection performance.

Domain adaptation methods aim to reduce dataset bias caused by a difference in the statistical distributions between training and test domains. In this paper, we treat the transformation of classifiers into detectors as a domain adaptation task. Many approaches have been proposed for classifier adaptation; e.g., feature space transformations [12], model adaptation approaches [13, 14] and joint feature and model adaptation [15, 16]. However, even the joint learning models are not able to modify the feature extraction process and so are limited to shallow adaptation techniques. Additionally, these methods only adapt between visual domains, keeping the task fixed, while we adapt both from a large visual domain to a smaller visual domain and from a classification task to a detection task.

Several supervised domain adaptation models have been proposed for object detection. Given a detector trained on a source domain, they adjust its parameters on labeled target domain data. These include variants for linear support vector machines [17, 18, 19], as well as adaptive latent SVMs [20] and adaptive exemplar SVM [21]. A related recent method [22] proposes a fast adaptation technique based on Linear Discriminant Analysis. These methods require labeled detection data for all object categories, both in the source and target domains, which is absent in our scenario. To our knowledge, ours is the first method to adapt to held-out categories that have no detection data.

## 3 Large Scale Detection through Adaptation (LSDA)

We propose Large Scale Detection through Adaptation (LSDA), an algorithm for adapting classifiers to detectors. With our algorithm, we are able to produce a detection network for all categories of interest, whether or not bounding boxes are available at training time (see Figure 2).

Suppose we have $K$ categories we want to detect, but we only have bounding box annotations for $m$ categories. We will refer to the set of categories with bounding box annotations as $B = \{1, ...m\}$, and the set of categories without bounding box annotations as set $A = \{m, ..., K\}$. In practice, we will likely have $m \ll K$, as is the case in the ImageNet dataset. We assume availability of classification data (image-level labels) for all $K$ categories and will use that data to initialize our network.

LSDA transforms image classifiers into object detectors using three key insights:

1. Recognizing background is an important step in adapting a classifier into a detector
2. Category invariant information can be transferred between the classifier and detector feature representations
3. There may be category specific differences between a classifier and a detector

We will next demonstrate how our method accomplishes each of these insights as we describe the training of LSDA.

## 3.1 Training LSDA: Category Invariant Adaptation

For our convolutional neural network, we adopt the architecture of Krizhevsky et al. [4], which achieved state-of-the-art performance on the ImageNet ILSVRC2012 classification challenge. Since this network requires a large amount of data and time to train its approximately 60 million parameters, we start by pre-training the CNN trained on the ILSVRC2012 classification dataset, which contains 1.2 million classification-labeled images of 1000 categories. Pre-training on this dataset has been shown to be a very effective technique [8, 5, 1], both in terms of performance and in terms of limiting the amount of in-domain labeled data needed to successfully tune the network. Next, we replace the last weight layer (1000 linear classifiers) with $K$ linear classifiers, one for each category in our task. This weight layer is randomly initialized and then we fine-tune the whole network on our classification data. At this point, we have a network that can take an image or a region proposal as input, and produce a set of scores for each of the $K$ categories. We find that even using the net trained on classification data in this way produces a strong baseline (see Section 4).

We next transform our classification network into a detection network. We do this by fine-tuning layers 1-7 using the available labeled detection data for categories in set $B$. Following the Regions-based CNN (R-CNN) [1] algorithm, we collect positive bounding boxes for each category in set $B$ as well as a set of background boxes using a region proposal algorithm, such as selective search [23]. We use each labeled region as a fine-tuning input to the CNN after padding and warping it to the CNN's input size. Note that the R-CNN fine-tuning algorithm requires bounding box annotated data for all categories and so can not directly be applied to train all $K$ detectors. Fine-tuning transforms all network weights (except for the linear classifiers for set $A$) and produces a softmax detector for categories in set $B$, which includes a weight vector for the new background class.

Layers 1-7 are shared between all categories in set $B$ and we find empirically that fine-tuning induces a generic, category invariant transformation of the classification network into a detection network. That is, even though fine-tuning sees no detection data for categories in set $A$, the network transforms in a way that automatically makes the original set $A$ image classifiers much more effective at detection (see Figure 3). Fine-tuning for detection also learns a background weight vector that encodes a generic "background" category. This background model is important for modeling the task shift from image classification, which does not include background distractors, to detection, which is dominated by background patches.

## 3.2 Training LSDA: Category Specific Adaptation

Finally, we learn a category specific transformation that will change the classifier model parameters into the detector model parameters that operate on the detection feature representation. The category specific output layer ($fc8$) is comprised of $fcA$, $fcB$, $\delta B$, and $fc - BG$. For categories in set $B$, this transformation can be learned through directly fine-tuning the category specific parameters $fc_B$ (Figure 2). This is equivalent to fixing $fc_B$ and learning a new layer, zero initialized, $\delta B$, with equivalent loss to $fc_B$, and adding together the outputs of $\delta B$ and $fc_B$.

Let us define the weights of the output layer of the original classification network as $W^c$, and the weights of the output layer of the adapted detection network as $W^d$. We know that for a category $i \in B$, the final detection weights should be computed as $W_i^d = W_i^c + \delta B_i$. However, since there is no detection data for categories in $A$, we can not directly learn a corresponding $\delta A$ layer during fine-tuning. Instead, we can approximate the fine-tuning that would have occurred to $fc_A$ had detection data been available. We do this by finding the nearest neighbors categories in set $B$ for each category in set $A$ and applying the average change. Here we define nearest neighbors as

those categories with the nearest (minimal Euclidean distance) $\ell_2$-normalized $fc_8$ parameters in the classification network. This corresponds to the classification model being most similar and hence, we assume, the detection model should be most similar. We denote the $k^{th}$ nearest neighbor in set $B$ of category $j \in A$ as $N_B(j, k)$, then we compute the final output detection weights for categories in set $A$ as:

$$\forall j \in A : W_j^d \quad = \quad W_j^c + \frac{1}{k} \sum_{i=1}^{k} \delta B_{N_B(j,i)} \tag{1}$$

Thus, we adapt the category specific parameters even without bounding boxes for categories in set $A$. In the next section we experiment with various values of $k$, including taking the full average: $k = |B|$.

### 3.3  Detection with LSDA

At test time we use our network to extract $K + 1$ scores per region proposal in an image (similar to the R-CNN [1] pipeline). One for each category and an additional score for the background category. Finally, for a given region, the score for category $i$ is computed by combining the per category score with the background score: $score_i - score_{background}$.

In contrast to the R-CNN [1] model which trains SVMs on the extracted features from layer 7 and bounding box regression on the extracted features from layer 5, we directly use the final score vector to produce the prediction scores without either of the retraining steps. This choice results in a small performance loss, but offers the flexibility of being able to directly combine the classification portion of the network that has no detection labeled data, and reduces the training time from 3 days to roughly 5.5 hours.

## 4  Experiments

To demonstrate the effectiveness of our approach we present quantitative results on the ILSVRC2013 detection dataset. The dataset offers a 200-category detection challenge. The training set has $\sim$400K annotated images and on average 1.534 object classes per image. The validation set has 20K annotated images with $\sim$50K annotated objects. We simulate having access to classification labels for all 200 categories and having detection annotations for only the first 100 categories (alphabetically sorted).

### 4.1  Experiment Setup & Implementation Details

We start by separating our data into classification and detection sets for training and a validation set for testing. Since the ILSVRC2013 training set has on average fewer objects per image than the validation set, we use this data as our classification data. To balance the categories we use $\approx$1000 images per class (200,000 total images). **Note**: for classification data we only have access to a single image-level annotation that gives a category label. In effect, since the training set may contain multiple objects, this single full-image label is a weak annotation, even compared to other classification training data sets. Next, we split the ILSVRC2013 validation set in half as [1] did, producing two sets: val1 and val2. To construct our detection training set, we take the images with bounding box labels from val1 for only the first 100 categories ($\approx$ 5000 images). Since the validation set is relatively small, we augment our detection set with 1000 bounding box annotated images per category from the ILSVRC2013 training set (following the protocol of [1]). Finally we use the second half of the ILSVRC2013 validation set (val2) for our evaluation.

We implemented our CNN architectures and execute all fine-tuning using the open source software package Caffe [24] and have made our model definitions weights publicly available.

### 4.2  Quantitative Analysis on Held-out Categories

We evaluate the importance of each component of our algorithm through an ablation study. As a baseline we consider training the network with only the classification data (no adaptation) and applying the network to the region proposals. The summary of the importance of our three adaptation components is shown in Figure 3. Our full LSDA model achieves a 50% relative mAP boost over

| Detection Adaptation Layers | Output Layer Adaptation | mAP Trained 100 Categories | mAP Held-out 100 Categories | mAP All 200 Categories |
|---|---|---|---|---|
| No Adapt (Classification Network) | | 12.63 | 10.31 | 11.90 |
| $\text{fc}_{bgrnd}$ | - | 14.93 | 12.22 | 13.60 |
| $\text{fc}_{bgrnd},\text{fc}_6$ | - | 24.72 | 13.72 | 19.20 |
| $\text{fc}_{bgrnd},\text{fc}_7$ | - | 23.41 | 14.57 | 19.00 |
| $\text{fc}_{bgrnd},\text{fc}_B$ | - | 18.04 | 11.74 | 14.90 |
| $\text{fc}_{bgrnd},\text{fc}_6,\text{fc}_7$ | - | 25.78 | 14.20 | 20.00 |
| $\text{fc}_{bgrnd},\text{fc}_6,\text{fc}_7,\text{fc}_B$ | - | 26.33 | 14.42 | 20.40 |
| $\text{fc}_{bgrnd}$,layers1-7,$\text{fc}_B$ | - | 27.81 | 15.85 | 21.83 |
| $\text{fc}_{bgrnd}$,layers1-7,$\text{fc}_B$ | Avg NN (k=5) | **28.12** | 15.97 | 22.05 |
| $\text{fc}_{bgrnd}$,layers1-7,$\text{fc}_B$ | Avg NN (k=10) | 27.95 | **16.15** | 22.05 |
| $\text{fc}_{bgrnd}$,layers1-7,$\text{fc}_B$ | Avg NN (k=100) | 27.91 | 15.96 | 21.94 |
| Oracle: Full Detection Network | | 29.72 | 26.25 | 28.00 |

Table 1: Ablation study for the components of LSDA. We consider removing different pieces of our algorithm to determine which pieces are essential. We consider training with the first 100 (alphabetically) categories of the ILSVRC2013 detection validation set (on val1) and report mean average precision (mAP) over the 100 trained on and 100 held out categories (on val2). We find the best improvement is from fine-tuning all layers and using category specific adaptation.

the classification only network. The most important step of our algorithm proved to be adapting the feature representation, while the least important was adapting the category specific parameter. This fits with our intuition that the main benefit of our approach is to transfer category invariant information from categories with known bounding box annotation to those without the bounding box annotations.

In Table 1, we present a more detailed analysis of the different adaptation techniques we could use to train the network. We find that the best category invariant adaptation approach is to learn the background category layer and adapt all convolutional and fully connected layers, bringing mAP on the held-out categories from 10.31% up to 15.85%. Additionally, using output layer adaptation ($k = 10$) further improves performance, bringing mAP to 16.15% on the held-out categories (statistically significant at $p = 0.017$ using a paired sample t-test [25]). The last row shows the performance achievable by our detection network if it had access to detection data for all 200 categories, and serves as a performance upper bound.[1]

We find that one of the biggest reasons our algorithm improves is from reducing localization error. For example, in Figure 4, we show that while the classification only trained net tends to focus on the most discriminative part of an object (ex: face of an animal) after our adaptation, we learn to localize the whole object (ex: entire body of the animal).

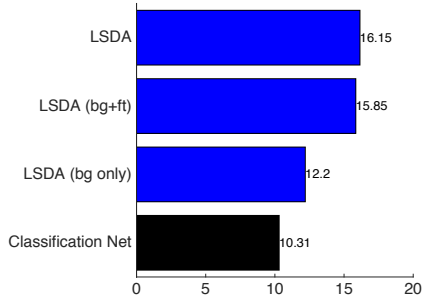

Figure 3: Comparison (mAP%) of our full system (LSDA) on categories with no bounding boxes at training time.

### 4.3 Error Analysis on Held Out Categories

We next present an analysis of the types of errors that our system (LSDA) makes on the held out object categories. First, in Figure 5, we consider three types of false positive errors: Loc (localization errors), BG (confusion with background), and Oth (other error types, which is essentially

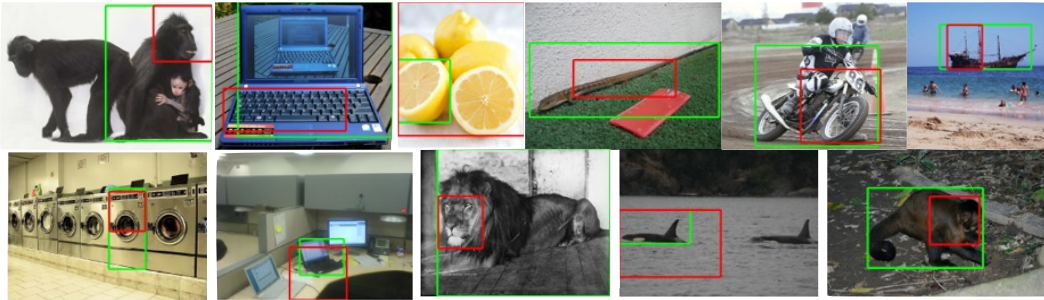

Figure 4: We show example detections on held out categories, for which we have **no detection training data**, where our adapted network (LSDA) (shown with green box) correctly localizes and labels the object of interest, while the classification network baseline (shown in red) incorrectly localizes the object. This demonstrates that our algorithm learns to adapt the classifier into a detector which is sensitive to localization and background rejection.

correctly localizing an object, but misclassifying it). After separating all false positives into one of these three error types we visually show the percentage of errors found in each type as you look at the top scoring 25-3200 false positives.[2] We consider the baseline of starting with the classification only network and show the false positive breakdown in Figure 5(b). Note that the majority of false positive errors are confusion with background and localization errors. In contrast, after adapting the network using LSDA we find that the errors found in the top false positives are far less due to localization and background confusion (see Figure 5(c)). Arguably one of the biggest differences between classification and detection is the ability to accurately localize objects and reject background. Therefore, we show that our method successfully adapts the classification parameters to be more suitable for detection.

In Figure 5(a) we show examples of the top scoring Oth error types for LSDA on the held-out categories. This means the detector localizes an incorrect object type. For example, the motorcycle detector localized and mislabeled bicycle and the lemon detector localized and mislabeled an orange. In general, we noticed that many of the top false positives from the Oth error type were confusion with very similar categories.

### 4.4 Large Scale Detection

To showcase the capabilities of our technique we produced a 7604 category detector. The first categories correspond to the 200 categories from the ILSVRC2013 challenge dataset which have bounding box labeled data available. The other 7404 categories correspond to leaf nodes in the ImageNet database and are trained using the available full image labeled classification data. We trained a full detection network using the 200 fully annotated categories and trained the other 7404 last layer nodes using only the classification data. Since we lack bounding box annotated data for the majority of the categories we show example top detections in Figure 6. The results are filtered using non-max suppression across categories to only show the highest scoring categories.

The main contribution of our algorithm is the adaptation technique for modifying a convolutional neural network for detection. However, the choice of network and how the net is used at test time both effect the detection time computation. We have therefore also implemented and released a version of our algorithm running with fast region proposals [27] on a spatial pyramid pooling network [28], reducing our detection time down to half a second per image (from 4s per image) with nearly the same performance. We hope that this will allow the use of our 7.6K model on large data sources such as videos. We have released the 7.6K model and code to run detection (both the way presented in this paper and our faster version) at `lsda.berkeleyvision.org`.

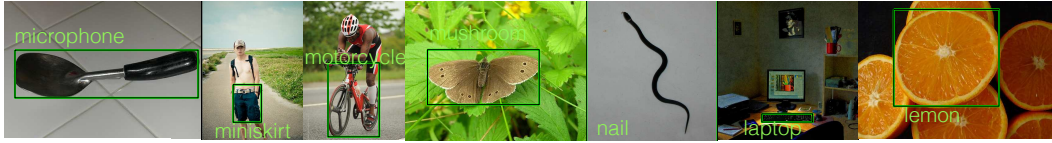

(a) Example Top Scoring False Positives: LSDA correctly localizes but incorrectly labels object

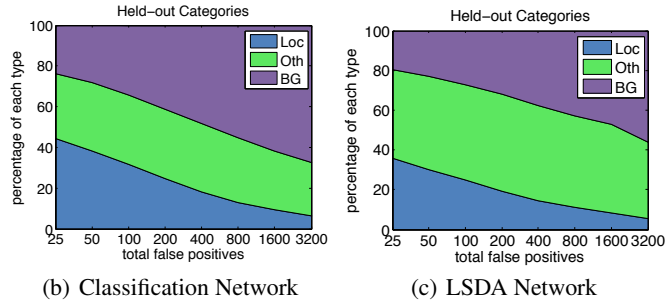

(b) Classification Network      (c) LSDA Network

Figure 5: We examine the top scoring false positives from LSDA. Many of our top scoring false positives come from confusion with other categories (a). (b-c) Comparison of error type breakdown on the categories which have no training bounding boxes available (held-out categories). After adapting the network using our algorithm (LSDA), the percentage of false positive errors due to localization and background confusion is reduced (c) as compared to directly using the classification network in a detection framework (b).

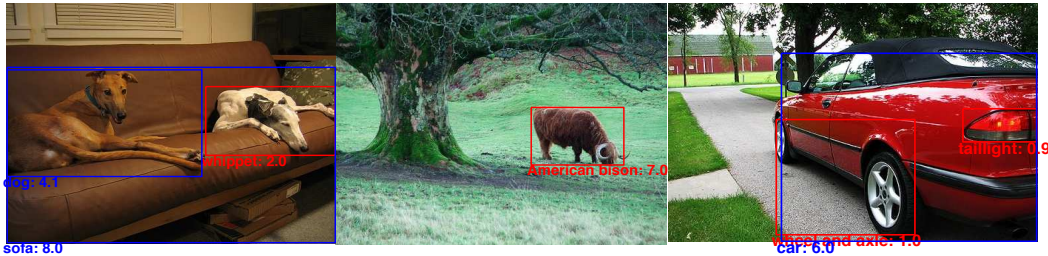

Figure 6: Example top detections from our 7604 category detector. Detections from the 200 categories that have bounding box training data available are shown in blue. Detections from the remaining 7404 categories for which only classification training data is available are shown in red.

## 5 Conclusion

We have presented an algorithm that is capable of transforming a classifier into a detector. We use CNN models to train both a classification and a detection network. Our multi-stage algorithm uses corresponding classification and detection data to learn the change from a classification CNN network to a detection CNN network, and applies that difference to future classifiers for which there is no available detection data.

We show quantitatively that without seeing any bounding box annotated data, we can increase performance of a classification network by 50% relative improvement using our adaptation algorithm. Given the significant improvement on the held out categories, our algorithm has the potential to enable detection of tens of thousands of categories. All that would be needed is to train a classification layer for the new categories and use our fine-tuned detection model along with our output layer adaptation techniques to update the classification parameters directly.

Our approach significantly reduces the overhead of producing a high quality detector. We hope that in doing so we will be able to minimize the gap between having strong large-scale classifiers and strong large-scale detectors. There is still a large gap to reach oracle (known bounding box labels) performance. For future work we would like to explore multiple instance learning techniques to discover and mine patches for the categories that lack bounding box data.

## Footnotes

[1]To achieve R-CNN performance requires additionally learning SVMs on the activations of layer 7 and bounding box regression on the activations of layer 5. Each of these steps adds between 1-2mAP at high computation cost and using the SVMs removes the adaptation capacity of the system.

[2]We modified the analysis software made available by Hoeim et al. [26] to work on ILSVRC-2013 detection

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
