[Reviews · NeurIPS 2014]

Submitted by Assigned_Reviewer_7

This paper addresses the issue of object detection, in particular the challenge of obtaining bounding boxes on a scale similar to which category labels exist for object categorization. The authors side-step this challenge by proposing to adapt object classifiers for the detection task. Their algorithm is fairly simple and straightforward, which is not a bad thing in itself. Their experimental protocol uses 100 categories for training (with both category labels and bounding boxes), and tests on 100 left-out categories. These left-out categories had only category labels. The results obtained were reasonably good, filling about 35% of the gap between an oracle and a baseline on the 100 left-out categories.

Quality and clarity: This is a good quality, clearly-written paper.

Originality: This paper combines two existing ideas related to adaptation: adapting object classifiers for detection, and adapting from one set of categories to another. To the best of this reviewer's knowledge, this is a novel and interesting idea -- but not ground-breaking.

Significance: The main contribution of this paper is the novel (but not ground-breaking) approach. Because results-wise, it is hard to judge the long-term significance, although they will likely be surpassed later by more sophisticated method. (This is to be expected). The main issue with the results is that they inevitably sacrifice performance (mAP) by using fewer bounding boxes, and there is no objective way to judge if this trade-off is worth it.

(As a side note, it is not clear if the issue being tackled is really an important long-term challenge. Yes, bounding boxes are more "expensive" to obtain than category labels. But who is to say that throwing resources at the problem won't fix it? Not too many years ago, having millions of images labeled with 20,000 object categories seemed almost impossible...)

Detailed issues that should be addressed:
- In Figure 3a, which methods do (or do not) use category labels to help with detection, other than their method and R-CNN? The authors should state this clearly, to be clear about an apples-to-apples comparison.
- My judgment of how good the proposed approach is, is based on the the blue bars in Figure 3b. DNN is about 1/3 between baseline and oracle. Do the authors agree that this is one of the main results? If yes, please highlight it, not the 78% (line 341) -- which is not quite the right number for 2 reasons: 1) should judge based only left-out categories, and 2) should not simply be a percentage of oracle performance (should take baseline into account). Analogous problem for the claim of 50% improvement (line 425).
- Authors should devote a table or a sub-section to clearly stating the differences from R-CNN, which seems to be the closest competing alternative. Currently, these statements seem to be scattered or not as prominent as they could be. I would suggest "sacrificing" Figure 6, which are simply anecdotal examples.

Minor issues (no need to respond)
- Is the proposed method called "DNN" or "DDA"???
- Typo on line 431 ("classi?ers")
Summary: Overall, this is a reasonably good paper proposing a reasonably novel approach. I believe the paper deserves to be accepted, and to be seen by the NIPS community. While the results will almost certainly be surpassed soon, since the method is simple (fine, this being the first paper to take this approach), the long-term significance of the paper (and the approach in particular) remains to be seen.

Submitted by Assigned_Reviewer_20

The authors proposed a framework to adapt deep neural networks trained from image classification tasks to object detection tasks. Evaluation on the ImageNet LSVRC-2013 dataset demonstrates that the proposed method ranks 4th in the objects detection challenge. This paper is clearly presented and generally easy to follow, and the proposed CNN structure is interesting and simple to implement. Over all, the quality of the paper is good. However, there are two concerns that the authors could improve.

First, as the main contribution of this work, the adaption method proposed in this paper needs better justification. The details of underlying motivations are not presented in a principled fashion. For example in Section 3.2 (Page 5 Line 231), it is not clear why the weights for adapted detection task is a summation of W_j^c and the W_{avg}. The physical meaning of this ‘offset parameter’ is not explained. As a consequence, it is also not clear why the kNN variant is better.

Second, in comparing with other methods, the users used part of the validation set, but not the test set, making the comparison not fair. The authors mentioned in footnote (Line 322) that RCNN has similar performance on val2 set, but due to the small size of this set, the performance gain is not statistically justified.
Summary: The authors proposed a framework to adapt deep neural networks trained from image classification tasks to object detection tasks. The proposed method has comparable performance as the state-of-the-art model, but the comparisons are on different settings, and the proposed adaptation method needs better explanation.

Submitted by Assigned_Reviewer_33

This paper presents a method for adapting a Krizhevsky CNN network that is trained for classification (without bounding box annotations) to work for object detection, given a small number of classes that have annotated bounding boxes. At train time, the approach 1) trains a CNN for classification on images of all classes, 2) trains a CNN on image regions on classes labeled with bounding boxes, while adding a negative background region class, and 3) adds the background class to the full CNN model while adapting some of the CNN weights. At test time, the CNN is run on multiple image regions in a test image.

Quality: This is an interesting paper and is fairly compelling that it gets reasonable results. These results are in large part building off the success of Krizhevsky et al.'s model, rather than something than something that is radically new; however, they will be valuable to researchers in computer vision and machine learning. The general approach is simple, but logical. Experiments are promising; however, it is missing comparisons to other baselines:
1) Use full images as positive examples, and images and regions from other classes as negative examples. Train and fine-tune all layers of the CNN.
2) Do the same as above, but use a MIL approach, where you iteratively select the highest scoring positive region for each image from set A (instead of the full image)
3) Train and fine-tune the CNN for classification. At test time, run the classifier and predict a single bounding box at the center of the image (e.g., the mean bounding box for ImageNet)

I am unsure as to whether or not the proposed approach would outperform these baselines. It isn't clear to me that the proposed approach makes more sense than baseline 1, and I think it probably makes less sense that baseline 2. Including the 3rd baseline would be important to convey to the reader how easy/hard the problem and dataset are.

Clarity: The paper is clear and well written. The clarity of the experiments section could be improved, in particular in terms elaborating on the explanation of what each method in table 1 refers to (does this mean applying Eq. 1 or 2 to a subset of CNN layers?)

Originality: The paper is sufficiently novel to be accepted, although the basic approach is simple and specific to certain types of CNN networks.

Significance: The problem and results are significant and valuable.
Summary: The paper addresses a relevant problem and the fact that decent detection results were obtained on a largescale dataset without bounding box labels is compelling. At the same time, the paper is missing some obvious baselines, and it is very possible that results are capitalizing off of improved results of CNN features rather than due to introducing a better way of training object detectors from weak supervision.
Author Feedback
Author rebuttal: Thank you to all the reviewers for your time and effort. Your feedback was very useful. We will begin by restating the main contribution of our paper. We present an algorithm that produces detectors for categories for which no bounding box annotations are available. We train a strong deep CNN detector on categories which have bounding box data, and then use that network to transfer information to the categories which lack bounding box annotations.This is a worthwhile goal because bounding boxes are not as easy to obtain compared to image-level labels. For example, since the submission deadline, we have already produced a detector for all leaf nodes in ImageNet (~7.5K categories), which have no bounding box annotations. This 7.5K detector utilizes the strong representations learned using all available bounding boxes in the 200-class ILSVRC13 challenge dataset. We stress that we do not propose to ignore available bounding boxes, we simply leave half of them out in the paper for evaluation purposes.

We would like to clarify our motivation for the adaptation of the output layer. Briefly, our adaptation method consists of taking the average change in the output layer weights observed during fine-tuning in set B, and adding it to the output layer weights in set A. For each category in set A, this average change is computed only for the most similar categories in set B. The motivation stems from the technique that is used to produce the fine-tuned weights, det-fcB. In particular, det-fcB is the result of running a back-propagation algorithm that is initialized with fcB (from the classification network). The backpropagation algorithm computes the error of the gradient and updates the weights of fcB by subtracting the weighted gradient and a momentum term. Therefore, after completing fine-tuning from classification to detection, we find that det-fcB = fcB - sum_i grad_i + momentum, or det-fcB = fcB + delta, where delta is the momentum minus gradients. Since the categories in set A have no bounding box annotations, fcA will not have any corresponding gradients computed during fine-tuning and so the weights won’t change. Therefore, we approximate how the weights **would have** changed during fine-tuning by taking the average gradient updates for the k-nearest neighbors for each category. If the paper is accepted, we will add more details of this motivation for the adaption of the output layer.

We include comments to specific reviewers below:

R20:
Note that val2 contains 10,000 images and is therefore a large set to evaluate our algorithm, in R-CNN paper authors claimed a good correspondence between results in val2 and in test. We plan to include our performance on the ILSVRC2013 test set once the server is online again. We have contacted the ImageNet organized and have been notified that the server will become available again on August 15th.

R33:
We appreciate your suggestions for extra comparison algorithms. We would be happy to compare against #1 and #3 in the final version of our paper. We agree that #3 would provide additional information about the difficulty of the dataset. Your suggestion #2 is an idea we are actively exploring, but we view it as an extension of our approach.

“It is very possible that results are capitalizing off of improved results of CNN features…” We would like to point out that we directly compare the same CNN architecture and test time detection pipeline using the classification weights vs our adapted detection weights with background. We argue that this shows that the improvement we observe (10.31->15.97) is not simply a result of using CNNs, but actually a result of our semi-supervised learning algorithm.

R7:
R-CNN, Overfeat, and NEC-MU use the ILSVRC-1000 dataset for pre-training a network. We will denote this on Fig3a. If accepted, we will also change the relative percentage improvement reported to be computed only on the held-out categories and for consistency we will change both relative computation references to be compared against the oracle detection network.